# Basin-Scale Sea Level Budget from Satellite Altimetry, Satellite Gravimetry, and Argo Data over 2005 to 2019

**Yuanyuan Yang [1,2], Wei Feng [3], Min Zhong [3], Dapeng Mu [4,* and Yanli Yao [1]**

1   School of City and Regional Planning, Yancheng Teachers University, Yancheng 224002, China
2   State Key Laboratory of Geodesy and Earth's Dynamics, Institute of Geodesy and Geophysics, Innovation Academy for Precision Measurement Science and Technology, Chinese Academy of Sciences, Wuhan 430077, China
3   School of Geospatial Engineering and Science, Sun Yat-Sen University, Zhuhai 519082, China
4   Institute of Space Sciences, Shandong University, Weihai 264209, China
*   Correspondence: mdp@sdu.edu.cn

**Abstract:** Monitoring sea level changes and exploring their causes are of great significance for future climate change predictions and the sustainable development of mankind. This study uses multiple sets of satellite altimetry, satellite gravity, and ocean temperature and salinity data to study the basin-scale sea level budget (SLB) from 2005 to 2019. The basin-scale sea level rises significantly at a rate of 2.48–4.31 mm/yr, for which the ocean mass component is a main and stable contributing factor, with a rate of 1.77–2.39 mm/yr, while the steric component explains a ~1 mm/yr sea level rise in most ocean basins, except for the Southern Ocean. Due to the drift in Argo salinity since 2016, the residuals of basin-scale SLB are significant from 2016 to 2019. The worst-affected ocean is the Atlantic Ocean, where the SLB is no longer closed from 2005 to 2019. If halosteric sea level change trends from 2005 to 2015 are used to revise salinity data after 2016, the SLB on the ocean basin scale can be kept closed. However, the SLB on the global scale is still not closed and requires further study. Therefore, we recommend that Argo salinity products after 2016 should be used with caution.

**Keywords:** sea level budget; salinity drift; GRACE; altimetry; Argo

## 1. Introduction

Sea level rise is one of the direct consequences of global warming. Monitoring sea level change and analyzing its causes are essential for understanding climate change [1–5]. Altimetry-based sea level change can be divided into two parts. One is ocean mass change, caused by the water transfer between ocean and land, and the other is steric sea level change, caused by the temperature and salinity variabilities in sea water [6–11]. With the development of satellite altimetry, satellite gravity, i.e., Gravity Recovery and Climate Experiment (GRACE), and the Argo oceanographic observation network, we can directly and accurately obtain these three quantities related to the sea level budget (SLB). Assuming the deep steric changes are negligible, the residual trend of global mean "Alt.–GRACE–Argo" is often less than its uncertainty, which is called the closure of the SLB [12–14]. The residual SLB can theoretically be used to explore the unknown deep ocean change, especially in the regional ocean basin scale, and assess possible systematic deviations between observations [15–18].

Previous studies have shown that the global- and basin-scale SLB are closed, with tolerance over different periods. Llovel et al. (2014) found that the residual trend of the SLB was −0.13 ± 0.72 mm/yr from 2005 to 2013 when considering the possible systematic uncertainties of altimetry, GRACE, and Argo data [19]. Comparing the observed sea level and the sum of its contributing factors since 1900, Frederikse et al. (2020) showed that the residual trends of the global SLB were 0.04 ± 0.36 mm/yr, 0.26 ± 0.33 mm/yr, and 0.19 ±

0.51 mm/yr for the periods of 1900–2018, 1957–2018, and 1993–2018, respectively. Moreover, the SLB of each ocean basin from 1993 to 2018 was also closed [20].

However, some studies have found significant differences in the global mean SLB since 2016. Royston et al. (2020) reported a significant discrepancy of about 1.2 mm/yr in the Indian Ocean–South Pacific basin for the SLB from 2005 to 2015, even when observation errors and systematic errors were considered [21]. Moreover, Chen et al. (2020) found that the global ocean mass change calculated from "Alt.–Argo" does not match that observed by GRACE and GRACE-FO since August 2016, which may be related to the abnormal operation of the accelerometer in the later period of GRACE and GRACE-FO era and the uncertainties of Argo and altimetry observations [22]. Barnoud et al. (2021) further confirmed that the salinity drift of Argo can explain about 40% of the non-closure of the global SLB from 2005 to 2019, while altimetry products have displayed good consistency since 2016 [23].

Is the ocean-basin-scale SLB closed since 2016? How does Argo salinity drift affect the regional SLB? To answer these questions, multiple sets of the latest altimetry, Argo, and GRACE/GRACE-FO data were adopted to carry out basin-scale SLB.

## 2. Data and Methodology

The data used here mainly include GRACE/GRACE-FO, satellite altimetry, temperature-salinity field products, and some auxiliary data.

Sea level anomaly grid datasets, retrieved from the Archiving, Validation, and Interpretation of Satellite Oceanographic (AVISO) and the Commonwealth Scientific and Industrial Research Organisation (CSIRO), are used to calculate the total sea level changes. Covering multiple satellite altimetry missions, the two products were standardized before release. In the joint calculation, the effects of glacial isostatic adjustment (GIA) and ocean bottom deformation (OBD) should also be corrected [24,25]. Two GIA models are used to correct gridded global sea level changes, while −0.3 mm/yr is subtracted for GIA correction on a global scale [26–28]. OBD corrections are calculated from spherical harmonic coefficient products released by GRACE/GRACE-FO [24,29].

To obtain the ocean mass change, the GRACE/GRACE-FO spherical harmonic solution and the mascon solution are considered in this study. Four sets of spherical harmonic coefficient gravity field products are used, which were released by the Center for Space Research (CSR) of the University of Texas at Austin, the German Research Centre for Geosciences (GFZ), the Jet Propulsion Laboratory (JPL) from NASA, and the Institute of Geodesy at Graz University of Technology (ITSG). We calculate the ocean bottom pressure (OBP) from the GSM and GAD data of GRACE products and then deduct the inverse barometer (IB) correction to invert ocean mass change [30]. During processing, low-order item replacement [27,31–33], GIA correction [26], and 300 km Gaussian filtering [34] need to be performed. In addition, CSR and JPL also provide grid-processed mascon products, which directly estimate the Earth's surface mass changes based on the inter-satellite variabilities of gravity satellites without complicated post-processing strategies [35,36]. In this paper, GRACE/GRACE-FO RL06 version data are used to calculate trend values, while the RL05 version is additionally included for uncertainty estimations.

Nine gridded temperature-salinity datasets are used to calculate the upper 2000 m steric sea level change, which is listed in Table 1 [11,37–44]. Most of them are based on the Argo Buoy Observation Network, and some also incorporate other ocean observation data such as CTD, XBT, and satellite remote sensing data. In addition to the discrepancy in the data source, the interpolation methods of their mapping are also different, e.g., gradual correction method in BOA, variational interpolation method in IPRC, and objective analysis method in NCEI. Therefore, it is recommended to synthesize multiple datasets to obtain a relatively reliable steric change [45,46].

**Table 1.** The Argo temperature and salinity field data used in this study.

| Index | Dataset | Horizontal Resolution | Vertical Resolution | Data Source | Reference |
|---|---|---|---|---|---|
| 1 | BOA | 1° × 1° | 0–1975 dbar, 58 layers | Argo | [37] |
| 2 | CORA | 1/2° (Mercator) | 1–2000 m, 152 layers | Argo + others | [38] |
| 3 | EN4_g10 | 1° × 1° | 5–5350 m, 42 layers | Argo + others | [39,40] |
| 4 | EN4_L09 | 1° × 1° | 5–5350 m, 42 layers | Argo + others | [39,41] |
| 5 | IAP | 1° × 1° | 0–2000 m, 41 layers | Argo + others | [42] |
| 6 | IPRC | 1° × 1° | 0–2000 m, 27 layers | Argo + others | http://apdrc.soest.hawaii.edu/ (accessed on 2 August 2022) |
| 7 | JAMSTEC | 1° × 1° | 10–2000 dbar, 25 layers | Argo + others | [43] |
| 8 | NCEI | 1° × 1° | 0–5500 m, 102 layers | Argo + others | [11] |
| 9 | SIO | 1° × 1° | 2.5–1975 dbar, 58 layers | Argo | [44] |

We follow the previous assumption that uncertainties in sea level change and its components are composed of mutually-independent observation errors and systematic errors [21]. Observational errors are characterized by the ensemble spread of data products released by different institutions, while systematic errors are bound by error laws combined with known error sources, i.e., errors related to the altimetry orbital altitude, OBD, GIA, and GRACE post-processing strategies [21,30,47]. Therefore, a final uncertainty estimate can be defined as the square root of the summation of trend standard error (s.e.), the observational error, and the systematic deviation. The final trend estimate is obtained from the ensemble mean of multiple datasets mentioned above. As shown in Table 2, for the uncertainty of altimetry, we mainly consider the uncertainties of the altimetry orbit and OBD given by Royston et al. (2020) and the ensemble spread of the three sets of altimetry products [21]. For the uncertainty of Argo, we only consider the uncertainty caused by a subjective selection of multiple sets of products. For the uncertainties of GRACE and GRACE-FO, we refer to the influence of the post-processing strategies recommended in the RL05 and RL06 for ocean mass estimation, which include GIA correction, C20, degree1 replacement, and the various filtering methods, i.e., 300 km Gaussian filtering, 500 km Gaussian filtering, Swenson de-striping, Chambers de-striping, and no filtering.

**Table 2.** Basin mean sea level trends and uncertainties from 2005 to 2019 (unit: mm/yr).

| | | Indian | | Atlantic | | Pacific | | Southern Ocean | | Global Ocean | |
|---|---|---|---|---|---|---|---|---|---|---|---|
| Altimetry | | | | | | | | | | | |
| Mean Trend | ±s.e. | 4.06 | 0.28 | 4.31 | 0.23 | 4.10 | 0.23 | 2.48 | 0.26 | 3.94 | 0.18 |
| | ensemble spread | | 0.67 | | 0.03 | | 0.03 | | 0.05 | | 0.12 |
| | orbital altitude | | 0.20 | | 0.22 | | 0.58 | | 0.13 | | 0.13 |
| | OBD | | 0.16 | | 0.16 | | 0.16 | | 0.16 | | 0.16 |
| | Quadratic sum of uncertainties | | 0.77 | | 0.36 | | 0.64 | | 0.34 | | 0.3 |
| Argo | | | | | | | | | | | |
| Mean Trend | ±s.e. | 1.63 | 0.29 | 1.27 | 0.12 | 1.01 | 0.12 | −0.03 | 0.13 | 1.05 | 0.08 |
| | ensemble spread | | 0.14 | | 0.12 | | 0.22 | | 0.22 | | 0.12 |
| | Quadratic sum of uncertainties | | 0.32 | | 0.17 | | 0.25 | | 0.26 | | 0.14 |
| GRACE | | | | | | | | | | | |
| Mean Trend | ±s.e. | 1.77 | 0.19 | 2.05 | 0.16 | 2.39 | 0.20 | 2.06 | 0.24 | 2.14 | 0.12 |
| | ensemble spread | | 0.17 | | 0.17 | | 0.05 | | 0.27 | | 0.04 |
| | degree1 spread | | 0.60 | | 0.22 | | 0.06 | | 1.05 | | 0.2 |
| | C20 spread | | 0.11 | | 0.08 | | 0.13 | | 0.37 | | 0.06 |
| | GIA spread | | 0.16 | | 0.03 | | 0.05 | | 0.30 | | 0.02 |
| | filter spread | | 0.06 | | 0.09 | | 0.02 | | 0.15 | | 0.03 |
| | Quadratic sum of uncertainties | | 0.68 | | 0.34 | | 0.26 | | 1.22 | | 0.25 |
| | Alt.–GRACE–Argo | 0.66 | 1.08 | 0.99 | 0.52 | 0.7 | 0.74 | 0.45 | 1.29 | 0.75 | 0.41 |

Considering that the Argo observation network did not have global coverage before 2005 and Argo products have not undergone strict post-processing since 2020, this study focuses on the period of January 2005 to December 2019. Since the spatial resolutions of each product varied, we interpolated these data linearly into 1° × 1° grids. Moreover, the regions where strong earthquakes have occurred (Sumatra-Andaman, Maule, and Tohoku-Oki earthquakes) are masked to avoid leakage effects from the possible co-seismic and post-seismic gravity changes on ocean mass estimations [48–50]. The coastal regions with 500 km buffers are excluded because of possible GRACE leakage errors and insufficient Argo sampling.

## 3. Results

### 3.1. Global and Basin-Scale Sea Level Budget

Figure 1 depicts the time series of global mean sea level change from 2005 to 2019, including the ocean mass change component $SSH_{mass}$, steric change component $SSL_{2000}$, and SLB Residual. Although the sea level rose or fell rapidly during El Niño and La Niña events, the global mean sea level showed an overall accelerated upward trend due to the intensification of global warming [51,52]. Increasing ocean mass and warming seawater are good explanations for current sea level rise. The SLB residual fluctuated around "0 mm" before 2016, which means that the SLB budget was closed from 2005 to 2015 [15,18]. However, the residuals have "significantly" increased since 2016, which seems a significant increasing trend. According to the statistics in Table 2, the rising rate of the global mean sea level was 3.94 ± 0.30 mm/yr from 2005 to 2019, for which the steric change and ocean mass change components contributed 1.05 ± 0.14 mm/yr and 2.14 ± 0.25 mm/yr, respectively. The residual trend of "Alt.–Argo–GRACE" was 0.75 ± 0.41 mm/yr from 2005 to 2019, which confirms that the global mean SLB was not closed [22].

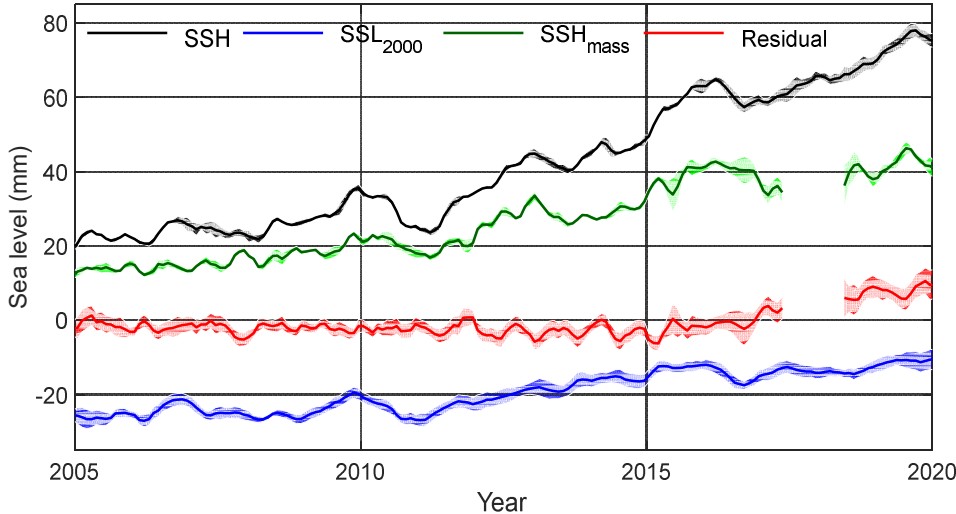

**Figure 1.** Time series of global mean sea level from 2005 to 2019 for altimetry-derived $SSH$ (black line), Argo-derived $SSL_{2000}$ (blue line), GRACE-derived $SSH_{mass}$ (dark green line), and residual (red line) data. The solid curves are the average of multiple products, and the shadings are corresponding standard deviations. Seasonal cycles were removed by least squares, and 3 months smoothing was applied for each series. Time series are offset for clarity.

Figure 2 shows the time series of basin mean sea level changes in the Indian Ocean, Atlantic Ocean, Pacific Ocean, and Southern Ocean. Compared with the global mean sea level (Figure 1), each basin sea level exhibits more short-term variability, particularly in the Indian Ocean. In addition to short time-scale changes, the sea level of the Indian Ocean

rose at a relatively uniform rate from 2005 to 2019, while the sea level of the other basins experienced similar acceleration or deceleration ascent processes to those of the global average. The series of ocean mass changes in each basin is in line with the total sea level change, which indicates that sea level changes are dominated by ocean mass change. However, the steric components in each basin are quite different, especially the obvious increase and decrease in the Indian Ocean sea level in 2006 and 2016, which may be related to short-time-scale climate events such as ENSO [53]. Moreover, the sea level residuals in each basin fluctuated around zero from 2005 to 2015 and were significantly increased by varying degrees after 2016, similar to those of the global averages.

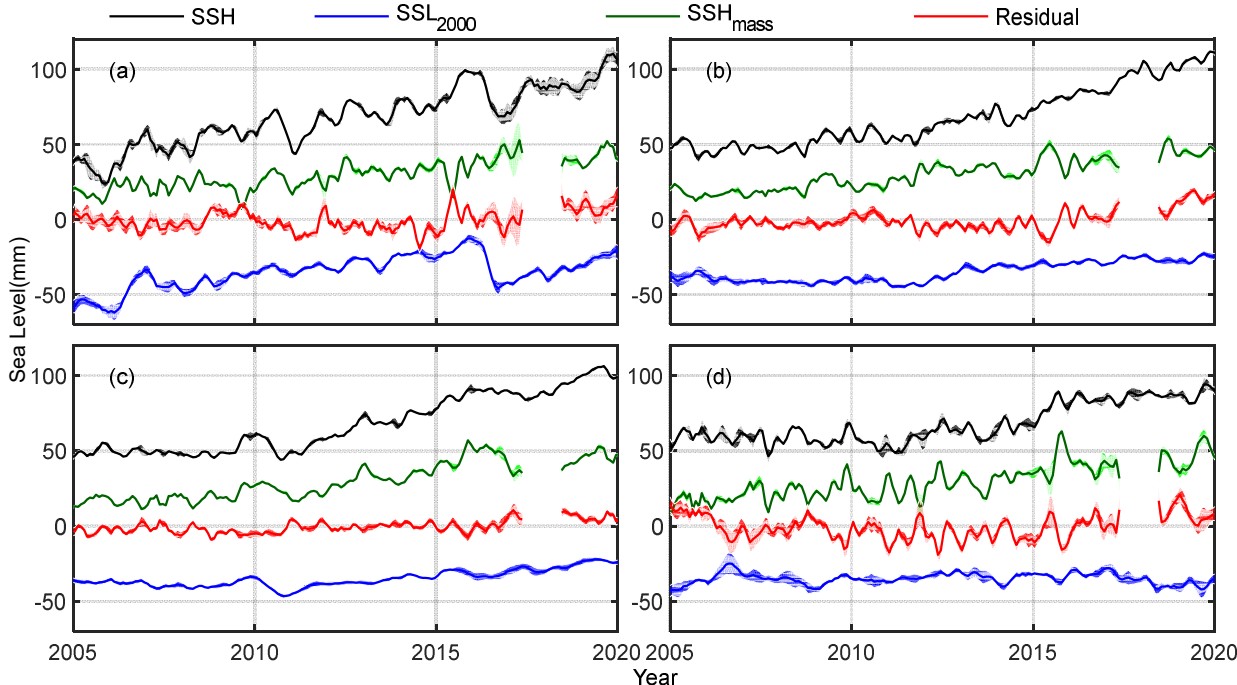

**Figure 2.** Time series of basin mean sea level change for (**a**) Indian Ocean, (**b**) Atlantic, (**c**) Pacific Ocean, and (**d**) Southern Ocean. The solid curves are averages of multiple products, and the shadings are corresponding standard deviations. Seasonal cycles were removed by least squares, and 3 months smoothing was applied for each series. Time series were offset for clarity.

Figure 3 depicts the statistics of the global mean and basin mean sea level trends from 2005 to 2019, including the products of various institutions and ensemble means. The ensemble spreads of altimetry-derived sea surface height products in these basins are relatively small, except for the Indian Ocean's. The discrepancies between GRACE products are mainly concentrated in the mascon and SHs solutions, especially in the Southern Ocean, where the impact of land-to-sea leakage errors is significant. The Argo products appear more discrete compared to the altimetry and GRACE products, so we should adopt as much data as possible to reduce possible errors. If we only consider the observation error between the datasets, the SLB in most oceans is unclosed, and the residual trends are not negligible compared to sea level components (Figure 3d).

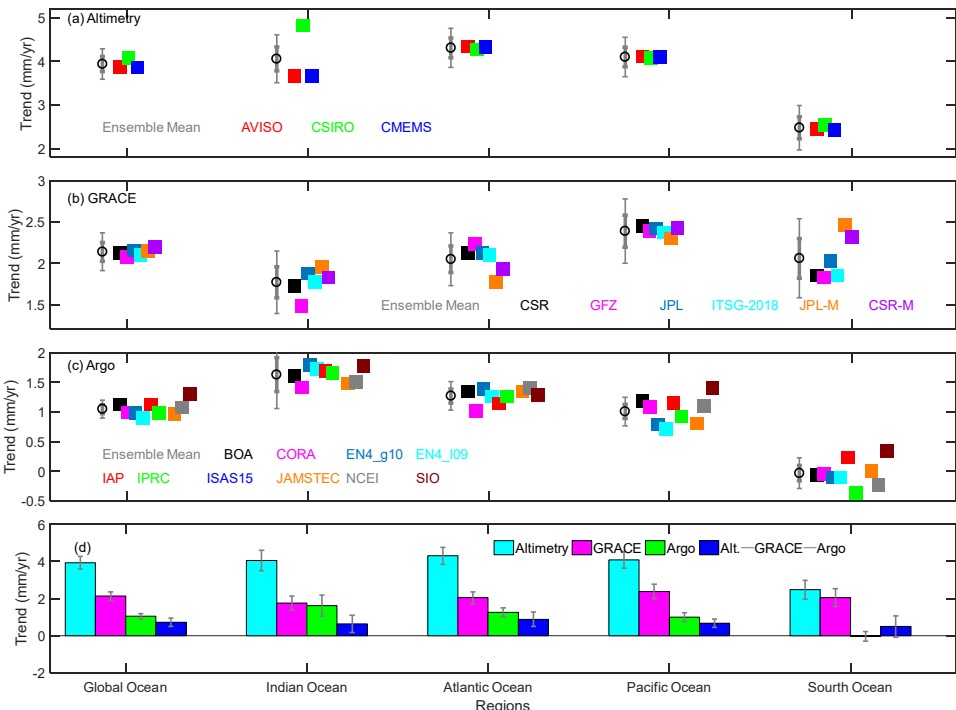

**Figure 3.** Global mean and basin mean sea level changes from 2005 to 2019, including trends of each dataset and ensemble averages. Trends of (**a**) total sea level changes derived from different altimetry datasets, (**b**) ocean mass changes derived from different GRACE solutions, (**c**) steric changes from different datasets, and (**d**) ensemble average trends. The error bar represents the error of the least squares linear fit (95% confidence interval).

Considering multiple datasets and comprehensive errors, Table 2 lists the trends and associated uncertainties of global- and basin-scale SLB. Except for the Southern Ocean, sea level rise rates in other ocean basins are slightly higher than the global average of 3.94 ± 0.30 mm/yr. In these basins, the largest contributing components to sea level rise are the ocean mass changes that are close to 2 mm/yr. However, the contribution of steric components to sea level rise varies greatly in different ocean basins. The highest rate of steric change is 1.63 ± 0.32 mm/yr in the Indian Ocean, the smallest is close to 0 in the Southern Ocean, and those of the other ocean basins are close to the global average. The SLB residual trends of the Indian Ocean, the Atlantic Ocean, the Pacific Ocean, and the Southern Ocean are 0.66 ± 1.08 mm/yr, 0.99 ± 0.52 mm/yr, 0.70 ± 0.74 mm/yr, and 0.45 ± 1.29 mm/yr, respectively. The SLB of the Atlantic Ocean is significantly not closed from 2005 to 2019, while that of the other basins is closed. Considering that the salinity drift was reported by the Argo Program Office (https://argo.ucsd.edu/, accessed on 2 August 2022), it is likely to be caused by the Argo salinity drift after 2016.

Recent basin-scale SLB results seem to be slightly different from the report of Royston et al. (2020) [21]. It should be noted that the statistical results we show here are based on the period of 2005–2019, which is different from their research period of 2005–2015. If our study period is restricted to the same span, we find that most basin-scale SLB is closed, consistent with the findings by Royston et al. (2020) [21]. Since the residual trend of the SLB is not uniform on a global scale, the closed state of the regional SLB also depends on the division of the ocean basin scale. Royston et al. (2020) merged the South Pacific and the North Indian Ocean together (Indian–South Pacific region) for analysis and found that the SLB is not closed in this region [21]. The explanation for this phenomenon is that the SLB is significantly unclosed in the North Indian Ocean and the South Pacific. At the basin

scale, the positive and negative signals of the SLB inside the ocean basin will cancel each other out and eventually reach a balance. In addition, subtle differences in numerical calculations, product selection, and uncertainty analyses will also have certain impacts on the results.

### 3.2. The Impact of Salinity Drift to Regional SLB

The steric sea level anomalies (SSLA) are mainly composed of two parts: thermosteric sea level anomalies (TSLA), caused by a thermal expansion or contraction of the ocean's volume, and the halosteric sea level anomalies (HSLA), derived from saline contraction or expansion of the ocean's volume [46,54–56]. We further studied the contribution of temperature and salinity to basin mean sea level rise, trying to understand the location and the impact of Argo salinity drift since 2016.

Figure 4 depicts the time series of HSLA, TSLA, and SSLA in each ocean basin. Temperature changes dominated the SSLA before 2016, and the contribution of salinity to sea level change was almost negligible. However, the Atlantic and Southern Oceans became significantly saltier after 2016, which led to regional sea level drops of 15 mm and 10 mm, respectively. This phenomenon conflicts with our general understanding that the ocean should be desalinated by the accelerated melting of glaciers and ice sheets in Antarctica and Greenland. This is most likely due to systematic deviation caused by the drift of Argo salinity data [57,58], which has a great impact on the basin-scale sea level budget, particularly in the Atlantic and Southern Oceans.

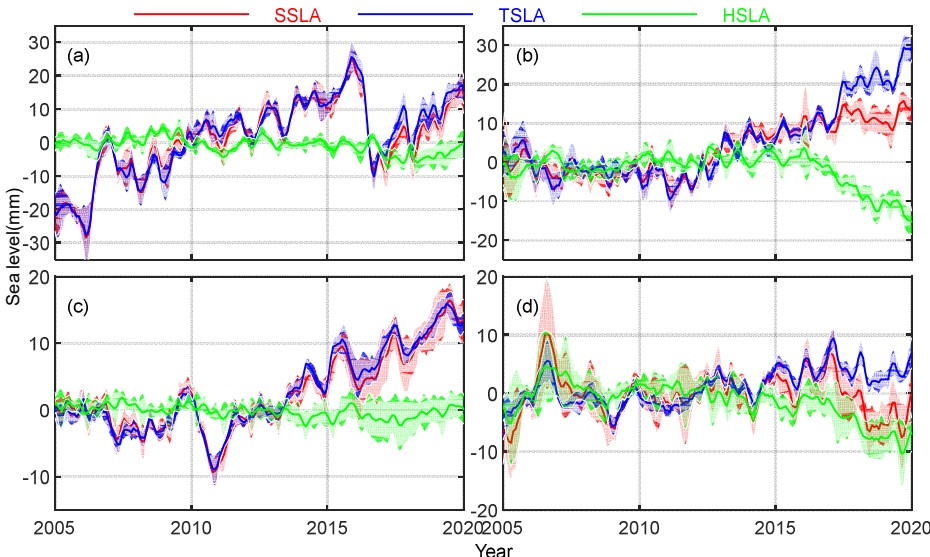

**Figure 4.** Time series of SSLA, TSLA, and HSLA for (**a**) Indian Ocean, (**b**) Atlantic Ocean, (**c**) Pacific Ocean, and (**d**) Southern Ocean.

Figure 5 illustrates the spatial distribution of SSLA, TSLA, and HSLA in the periods of 2005–2019, 2005–2015, and 2016–2019. The spatial distribution characteristics of steric change and its components in 2005–2019 are similar to those in 2005–2015. The steric sea levels are decreasing significantly in the Northeast Atlantic Ocean and the Subtropical Western Pacific Ocean while increasing in the Northwest Atlantic Ocean, the Equatorial East Pacific Ocean, and the East Indian Ocean over the periods of 2005–2015 and 2005–2019. Since ocean temperature changes contribute to most of the steric sea level changes, the patterns of SSLA and TSLA are roughly the same [59]. In addition, HSLA changes are relatively weak in most of the ocean, but we cannot ignore the compensatory effect of salinity in the North Atlantic and the gaining effect in the East Indian Ocean [45,46,60].

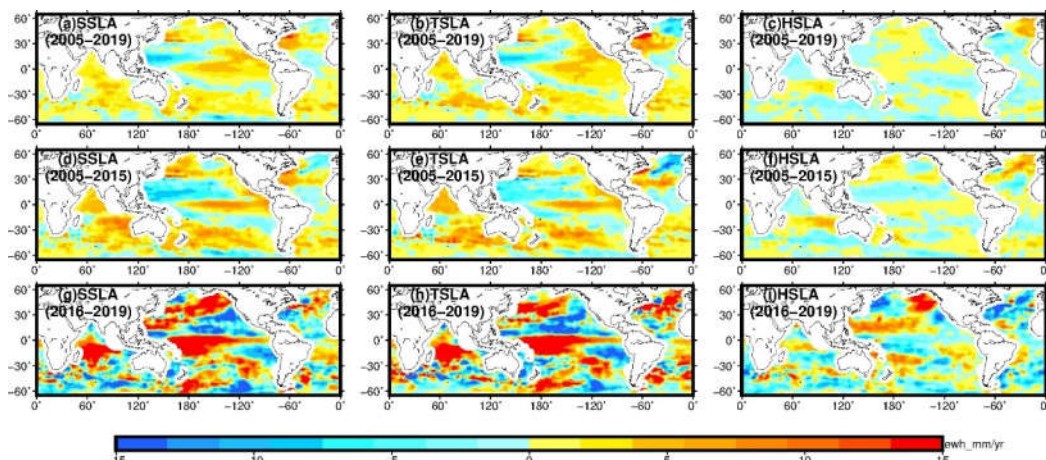

**Figure 5.** Comparison trend patterns of SSLA, TSLA, and HSLA over 2005–2019, 2005–2015, and 2016–2019.

The trend signals for 2016–2019 significantly exceed those of 2005–2015 and 2005–2019, which means that the short-time-scale changes may be more severe. In addition to the trend amplitudes, we can find that the steric trends in some basins around 2016 are completely opposite, especially in most of the North Atlantic basin. The SSLA trend signals of the Northwest Atlantic change from positive to negative, while those of the Northeast Atlantic change from negative to positive. The opposing changes of steric trends in the North Atlantic around 2016 are caused by the combined effects of temperature and salinity. There are positive and negative patterns of TSLA in the North Atlantic from 2005 to 2015, but almost exclusively positive signals in 2016–2019. The spatial distribution of HSLA in the North Atlantic shows a change from positive to negative signals.

In addition, the salinity changes from 2016 to 2019 in the eastern Indian Ocean, most of the Pacific Ocean, and the southern Atlantic Ocean close to the westerly drift zone are diametrically opposite to those of 2005–2015 (Figure 5f,i). Since the positive and negative signals of salinity changes cancel each other out after the weighted average of the basin scale, there are no clear trends for HSLA series of the Indian Ocean and the Pacific Ocean (Figure 4a,c). Due to the influence of the drift of salinity data since 2016, we are temporarily unable to determine whether Figure 5i is credible, unless there is definite confirmation from third-party data.

Table 3 shows the basin mean trends of HSLA in two time periods. During 2005 to 2015, we can infer that the contribution of salinity to sea level rise was very weak, or even negligible, even in regional oceans. Due to the Argo salinity drift, HSLA showed a significant negative trend from 2005 to 2019, which had a great impact on the basin-scale SLB, i.e., 0.76 mm/yr in the Atlantic Ocean and 0.3 mm/yr in the Southern Ocean. Assuming the basin mean HSLA trends from 2005 to 2015 are correct, we adopted these values in the current period to revise the SLB residual trends. The corrected residual trends are 0.49 ± 1.08 mm/yr, 0.23 ± 0.52 mm/yr, 0.65 ± 0.74 mm/yr, and 0 ± 1.29 mm/yr for the Indian, Atlantic, Pacific, and Southern Oceans, respectively. Obviously, this method is simple and effective to close the basin-scale SLB.

However, only considering the salinity drift is not enough to close the global SLB from 2005 to 2019 (0.55 ± 0.41 mm/yr), which has been reported by Barnoud et al. (2021) [23]. It seems to be in contradiction with the closure of the SLB at each basin. One possible reason is that the global average conceals regional differences, which in turn leads to an underestimation of uncertainty on a global scale. Another possibility is a potential systematic deviation of ocean mass estimations from GRACE and GRACE-FO [23]. In addition, the error of the numerical calculations cannot be ruled out because the global SLB residual trend and the uncertainty after a salinity drift correction are very close.

**Table 3.** Basin mean trend of HSLA (unit: mm/yr).

| Time | Indian | Atlantic | Pacific | Southern Ocean | Global Ocean |
|---|---|---|---|---|---|
| HSLA (2005–2015) | −0.13 ± 0.18 | 0.29 ± 0.17 | −0.15 ± 0.06 | −0.21 ± 0.21 | −0.05 ± 0.07 |
| HSLA (2005–2019) | −0.30 ± 0.12 | −0.47 ± 0.26 | −0.10 ± 0.04 | −0.51 ± 0.16 | −0.25 ± 0.07 |
| Salinity drift | 0.17 | 0.76 | 0.05 | 0.3 | 0.2 |
| Revised residual trends from 2005 to 2019 | 0.49 ± 1.08 | 0.23 ± 0.52 | 0.65 ± 0.74 | 0.15 ± 1.29 | 0.55 ± 0.41 |

## 4. Conclusions

Multiple sets of satellite altimetry, satellite gravity, and ocean temperature and salinity data were used to study the global mean and basin mean SLB from 2005 to 2019. The basin mean sea level rose significantly, with rates ranging from 2.48 ± 0.34 mm/yr to 4.31 ± 0.36 mm/yr, compared with a global mean rate of 3.94 ± 0.3 mm/yr. The increase in ocean mass is the main contributor to sea level rise in all basins, with rates ranging from 1.77 ± 0.68 mm/yr to 2.39 ± 0.26 mm/yr. The increases in steric sea levels in the Indian Ocean, the Atlantic Ocean, and the Pacific Ocean also contributed more to the sea level rises, with rates of more than 1 mm/yr, while the steric change in the Southern Ocean was not significant.

After comprehensively considering observational errors and systematic biases, we found that the SLB in the Atlantic was significantly unclosed from 2005 to 2019, with a trend of 0.99 ± 0.52 mm/yr. According to the time series and spatial distribution of HSLA, the systematic deviation of the Argo salinity data since 2016 is the main reason for the non-closure of the Atlantic SLB. Due to the salinity drift, HSLA of the ocean basins changed drastically around 2016. The basins most affected by salinity drift are the Atlantic Ocean and the Southern Ocean, both of which show systematic negative trends in HSLA after 2016. Given that the salinity drift has affected the global and basin scale SLB, we should be cautious when using the salinity data after 2016.

A simple and effective hypothesis that the contribution of salinity to sea level change after 2016 is consistent with that of 2005–2015 was adopted to revise the SLB. The revised SLB of each ocean basin is closed and the residual trends are significantly reduced. However, the global mean SLB still cannot be closed, which may be related to GRACE/GRACE-FO data, numerical calculations, and the error estimation method. Two more rigorous approaches are using ship-based high-precision salinity observations to calibrate Argo observations and quantifying the salinity changes from the perspective of the global water cycle, which will also be the focus of future work.

**Author Contributions:** Data curation, Y.Y. (Yanli Yao); Software, W.F.; Supervision, M.Z.; Writing—original draft, Y.Y. (Yuanyuan Yang); Writing—review & editing, D.M., W.F. All authors have read and agreed to the published version of the manuscript.

**Funding:** This research was funded by the National Natural Science Foundation of China (41904081 and 42192534), the State Key Laboratory of Geodesy and Earth's Dynamics, Chinese Academy of Sciences (SKLGED2022-2-1), the Natural Science Fund for Distinguished Young Scholars of Hubei Province, China (2019CFA091), and the Fundamental Research Funds for the Central Universities, Sun Yat-sen University (22lgqb09).

**Acknowledgments:** Thanks to the constructive comments given by three anonymous reviewers, which are very helpful for the improvement of the manuscript. Thanks to the editor for the APC discount. GRACE and GRACE-FO solutions provided by CSR, JPL, GFZ, and ITSG can be downloaded from the website http://icgem.gfz-potsdam.de/ (accessed on 2 August 2022). The temperature and salinity data used in this paper can be obtained from the reference or the website in Table 1. Satellite altimetry data can be obtained from the websites https://www.aviso.altimetry.fr/ (accessed on 2 August 2022), https://marine.copernicus.eu/ (accessed on 2 August 2022), and http://www.cmar.csiro.au/sealevel/sl_data_cmar.html (accessed on 2 August 2022).

**Conflicts of Interest:** The authors declare no conflicts of interest.

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
