# Peer review of "Basin-Scale Sea Level Budget from Satellite Altimetry, Satellite Gravimetry, and Argo Data over 2005 to 2019"

_remotesensing, doi:10.3390/rs14184637_

Round 1

Reviewer 1 Report

In the paper "Basin-Scale Sea Level Budget from altimetry, Argo and GRACE data over 2005 to 2019 " the authors present used to study the global-mean and basin-mean sea level budget from 2005 to 2019 by using multiple data sets: satellite altimetry, satellite gravity, and ocean temperature and salinity.

Comments:
1. The Abstract should be definitely improved, first of all missing the goal of the article is missing. The objective of this manuscript is clearly stated at the end, not in the abstract.
2. The Methodology section must be completed and please explain the methods in detail so that the results can be well understood.
3. I missing the Discussion section.  Furthermore, a more effective Discussion should be performed as some of the Conclusions had not been analysed previously.
4. Most illustrations are in good shape.
5. "This article is structured as follows. Section 2 briefly describes the data and methodology. Section 3 introduces basin-mean sea level budget and analyses the reasons for the unclosed sea level budget since 2016. Section 4 is the summary of the paper. " - I think, it is an unnecessary sentence in the Abstract section.
5. How had the Seasonal cycles been removed?

Overall, this is an interesting manuscript. There is certainly value in research. However, in the current state, I cannot recommend the publication of this manuscript, should be revised before another submittal.

Reviewer 2 Report

Review of "Basin-Scale Sea Level Budget from altimetry, Argo and GRACE 2

data over 2005 to 2019 (RS-1872140)”

Aug 21, 2022

Summary

This manuscript illustrated the basin-scale sea level budget from multiple products of altimetry, Argo and GRACE over 2005 to 2019. They found the halosteric data with considerable systematic error contributed to the non-closure sea level since 2016. The reviewer recommends a major revision along with substantial English editing and details representing before getting it published. Also, the table information and result are not clear as to compare with your salinity drift correction to existing papers. Please find my major and minor comments as follows:

Major Comments

(1) I feel the revision has many readability issue in the introduction, data, and methodology sections, which needs improvement. For example, “Altimetry-based sea level change can be divided into two parts,(:) one is …, and the other is ….” and “Moreover, the sea level budget of each ocean basin from 1993 to 2018 45 is (was) also closed [10].” … Please check your grammar carefully.

Also, the “..water transfer of ocean and land..” should be “..water transfer between ocean and land”.

(2) “In the joint calculation, the effects of glacial isostatic adjustment (GIA) and ocean bottom deformation (OBD) should also be corrected. Two GIA models are used to correct gridded global sea level changes [13,14], while -0.3 mm/yr is subtracted for GIA correction on a global scale.” Why should these corrections be applied to these products? It would be more readable if you give reasons behind these processes.

(3) Line140-142 “The residual trend of “Alt.-Argo-GRACE” is 0.75±0.41mm/yr from 2005 to 2019, which confirms that the global mean sea level budget is not closed [12].” Is there any standard to quantitatively determine whether the sea level is closed or not? If there is, please clarify because you have given many references showing different numerical sea level budget without clear comparison.

(4) Line 242-244 “Considering that the salinity change is a very long process and the salinity drift reported by the Argo Pro-gram Office (https://argo.ucsd.edu/), we think Figure 5(i) is likely to be unreliable.” This interpretation is not convincing. What do you mean “a very long process”? Please further clarify since this is the main finding of the study.

(5) Line 245-Line 254 “Due to the Argo salinity drift, HSLA showed a false negative trend from 2005 to 2019, which has a great impact on the basin-scale SLB, i.e., 0.76 mm/yr in the Atlantic Ocean and 0.3 mm/yr in the Southern Ocean.” It is dangerous to say “false” results because the study results remain controversial.

“Assuming the basin-mean HSLA trends from 2005 to 2015 are correct, we adopt these values in the current period to revise the SLB residual trends.” This is highly subjective assumption and so is the process of applying past results to the “current period”, even if you have mentioned the probable weakness of this method in Line 255-262. This method might be simple, but could also be “false” because you did not compare with results from other methods to validate yours. Please provide more comparisons or other firm references to support your method.

Minor Comments

Introduction

1. Line 30-33, “With the development of satellite gravity, i.e., Gravity Recovery and Climate Experiment (GRACE) and Argo oceanographic observation network, we can directly and accurately obtain these three quantities related to the sea level budget.” Please add up the altimetry product.

2. Paragraph 3 needs a central sentence.

Data and Methodology

3. “… and the auxiliary date in these data processing.” What is an “auxiliary date”? Perhaps you mean “the study period”.

4. “According to the law of error propagation, … OBD, GIA, and GRACE post-processing strategies [11,17,24].” Please simplify these sentences.

5. There are some confusions on Line 103-118 and the Table 2. What is the difference between ensemble spread and quadratic sum of uncertainties? What is ‘s.e.’? Please carefully rewrite Line 103-118 to clarify these terms shown in the table to avoid misleading information.

6. “The errors of GIA correction, …  recommended strategies of GRACE RL05 and RL06.” What do you mean “the spread” of these filtering methods? the average? Besides, which release (05 or 06) of GRACE did you particularly use in this study? Please clarify.

Results

1. L132-135 “Except for seasonal and interannual …  accelerate in 2011, and slow increase after 2016 [28,30].” Please rewrite this sentence.

2. L135-136 “The SLB residual fluctuated around “0” before 2016, which means that the SLB budget was closed from 2005 to 2015 [5,8].” It should be “0 mm”.

3. You should put Table 2 and Figure 3 forward in section 3.1. It is hard for readers to connect the descriptions to the figure and table in context. Besides, please explain the negative SSL in Figure 1.

4. Line 144-146 “Compared with the global mean sea level (Figure 1), each basin exhibits more sea level fluctuations, especially the Indian Ocean.” Please rewrite this sentence.

5. Line 148 “… a similar acceleration and deceleration ascent …”  It should be “or” rather than “and”.

6. Line 151 Be careful of the word “especially” and its usage in the construction of sentences. Since you have used it a lot of times, please check.

7. Line 203-205. You need to define the “… halosteric sea-level anomalies (HSLA), thermosteric sea-level anomalies (TSLA)… ” before using them in your results.

8. Line 218 “… over two time periods” Please specify the two time periods.

9. It is hard to recognize the “North Atlantic” in Line 230 or “southern Atlantic Ocean” in Line 238 or “Southern Ocean” in Line 249. Please check your description of places and make sure they are uniform.

10. Line 242 “… positive and negative signals …  cancel ….” What do you mean “cancel”? Perhaps you mean “disappear”.

11. Line 242-244 “Considering that the salinity change is a very long process and the salinity drift reported by the Argo Program Office (https://argo.ucsd.edu/), we think Figure 5(i) is likely to be unreliable.” This interpretation is not convincing to me. What do you mean “a very long process”? Please further clarify.

Reviewer 3 Report

This paper studied the sea level budget using satellite data, Argo network. and GRACE data. I believe the subject is important since is one of main indicators of the climate changes.  However, the authors presented a clear and easy methodology to show their recommendation for studying SLB. The results are well presented as well as the conclusion. 

I am afraid that the papers doesn't contain enough references related to subject and more discussion of the results. 

I agree for the publication of this paper if the authors improve it after a minor revision, taking into consideration these remarks and others in the document attached. 

Round 2

Reviewer 1 Report

The authors significantly improved the manuscript and replied to all the comments from the reviewers. Most of the required corrections were implemented. I recommend this manuscript for publication.

Reviewer 2 Report

The manuscript has been significantly improved. It can be accepted in present form.